# Effect of exogenous alpha-tocopherol on physio-biochemical attributes and agronomic performance of lentil (*Lens culinaris* Medik.) under drought stress

**Wadood Shah[1], Sami Ullah[1]\*, Sajjad Ali[2], Muhammad Idrees[3], Muhammad Nauman Khan[2], Kashif Ali[4], Ajmal Khan[5], Muhammad Ali[5], Farhan Younas[6]**

1 Department of Botany, University of Peshawar, Peshawar, Pakistan, 2 Department of Botany, Bacha Khan University, Charsadda, Pakistan, 3 Department of Chemistry, Bacha Khan University, Charsadda, Pakistan, 4 School of Ecology and Environmental Science, Yunnan University, Kunming, China, 5 Department of Biotechnology, Bacha Khan University, Charsadda, Pakistan, 6 Centre of Interdisciplinary Research in Basic Sciences, International Islamic University, Islamabad, Pakistan

\* sami_jan69@yahoo.com

**Data Availability Statement:** All relevant data are within the manuscript.

## Abstract

Water being a vital part of cell protoplasm plays a significant role in sustaining life on earth; however, drastic changes in climatic conditions lead to limiting the availability of water and causing other environmental adversities. α-tocopherol being a powerful antioxidant, protects lipid membranes from the drastic effects of oxidative stress by deactivating singlet oxygen, reducing superoxide radicals, and terminating lipid peroxidation by reducing fatty acyl per-oxy radicals under drought stress conditions. A pot experiment was conducted and two groups of lentil cultivar (Punjab-2009) were exposed to 20 and 25 days of drought induced stress by restricting the availability of water after $60^{th}$ day of germination. Both of the groups were sprinkled with α-tocopherol 100, 200 and 300 mg/L. Induced water deficit stress conditions caused a pronounced decline in growth parameters including absolute growth rate (AGR), leaf area index (LAI), leaf area ratio (LAR), root shoot ratio (RSR), relative growth rate (RGR), chlorophyll a, b, total chlorophyll content, carotenoids, and soluble protein content (SPC) which were significantly enhanced by exogenously applied α-tocopherol. Moreover, a significant increase was reported in total proline content (TPC), soluble sugar content (SSC), glycine betaine (GB) content, endogenous tocopherol levels, ascorbate peroxidase (APX), catalase (CAT) peroxidase (POD) and superoxide dismutase (SOD) activities. On the contrary, exogenously applied α-tocopherol significantly reduced the concentrations of malondialdehyde (MDA) and hydrogen peroxide ($H_2O_2$). In conclusion, it was confirmed that exogenous application of α-tocopherol under drought induced stress regimes resulted in membrane protection by inhibiting lipid peroxidation, enhancing the activities of antioxidative enzymes (APX, CAT, POD, and SOD) and accumulation of osmolytes such as glycine betaine, proline and sugar. Consequently, modulating different growth, physiological and biochemical attributes.

**Funding:** No funding was given by any source to conduct this study but this study is a part of M. Phil/MS degree of Mr. Wadood Shah for which Botany Department, University of Peshawar, Pakistan provided the laboratory facilities.

**Competing interests:** The authors have declared that no competing interests exist.

**Abbreviations:** AGR, Absolute growth rate; APX, Ascorbate peroxidase; α, Alpha; CAT, Catalase; $CO_2$, Carbon dioxide; GB, Glycine betaine; $H_2O_2$, Hydrogen peroxide; LAI, Leaf area index; LAR, Leaf area ratio; MDA, Malondialdehyde; POD, Peroxidase; ROS, Reactive oxygen species; RGR, Relative growth rate; RSR, Root shoot ratio; SOD, Superoxide dismutase; SPC, Soluble protein content; SSC, Soluble sugar content; TPC, Total proline content; CGR, Crop growth rate; NAR, net assimilation rate; CVG, Coefficient of velocity of germination; RWC, relative water content; SVI, seed vigor index.

## 1. Introduction

Changing climatic condition is becoming an obstacle in fulfilling the demand of food and achieving a sustainable agriculture; climatic changes result in droughts, heavy floods, earthquakes, fluctuation in temperature and other environmental adversities, that ultimately lead to reduce crop productivity [1]. Amongst major abiotic stresses, drought stress has a profound effect on crop growth and yield reduction; though, plants can often withstand limited water condition but at the cost of substantial amount of plant total biomass and yield loss, drought stress condition disturbs vital physiological and biochemical processes ultimately declining plant growth and production [2]. About 50% of the world's arid and semi-arid regions are being exposed to some kind of drought stresses [3]. With increasing climatic changes crops are losing their yield potential, thus making it hard to fulfil the increasing demand of food around the world [4]. The population of Pakistan is rapidly increasing with a growth rate of 2.1%, which is higher than the growth rate (1.1%) of world population. Keeping in mind the present scenario of climate change, it is predicted that 2.1 million hectares of land of Pakistan will be affected by drought by 2025 [5]. In general, during the spring season crops of Punjab suffer from drought stress due to higher rate of transpiration and elevated temperature. Most parts of the central, southern Punjab and parts of eastern Sindh do not fall under the domain of winter rains, and 50% of the time remains dry, thus considered drought susceptible zones [6].

Abiotic stresses affect photosynthesis, cell growth, development and other vital physiological and biochemical processes [7]. According to previous research findings, it is evident that water shortage in plants prompts oxidative stress in the forms of free radicals and non-radicals; both of the oxidants produced in response to abiotic stress damage biological membranes and other essential biomolecules such as proteins, lipids, chlorophyll and DNA [8]. At the onset of stress conditions, plants tend to accumulate various types of osmolytes such as sugars, proteins, proline and glycine betaine. Osmolytes chiefly accumulate in the cytoplasm preventing cellular degradation and maintain osmoregulation. Owing to their non-toxic nature and high solubility they do not impede other physio-biochemical processes [9].

Drought stress tolerance can be enhanced by using latest techniques like genetic engineering and tissue culture; however, these methods are expensive and inflict long term health concerns.α-tocopherol being a powerful antioxidant, attenuating the negative impacts of drought stress by scavenging free radicals and inhibiting lipid peroxidation, thus shielding biological membranes against oxidative stress. It is suggested that one molecule of a-tocopherol can deactivate 120 molecules of $^1O_2$ under drought stress condition [10]. After encountering drought stress regimes, plants trigger powerful antioxidant systems in the form of vitamins, flavonoids, carotenoids and antioxidant enzymes mainly including peroxidase, catalase, superoxide dismutase, glutathione reductase, and ascorbate peroxidase [11].

Lentil (*Lens culinaris* Medik.) is annual self-pollinated specie belonging to family *Leguminacae* (*Fabaceae*). It is widely grown in South Asia, Middle East, North America, North Africa, and Australia. Protein content of lentil seed ranges from 22% to 34.6% making it the third highest level of protein of any legume or nut after soybeans and hemp. Lentil is one of the major cash crops of Pakistan and it is extensively grown in Punjab and Khyber pakhtunkhwa province of Pakistan [12].

The present research work was aimed to assess growth, physiological and biochemical responses of lentil cultivar (Punjab-2009) to varying levels of exogenously applied α-tocopherol, its potential in alleviating the negative impacts of drought induced stress and to explore the degree of efficacy of α-tocopherol in regulating vital metabolic processes by ameliorating drought tolerance potential of lentil cultivar subjected to varying levels of drought induced stress.

## 2. Materials and methods

### 2.1. Site description and experimental design

Field experiment was carried out at the Department of Botany, University of Peshawar (34˚ 1' 33.3012" N and 71˚ 33' 36.4860" E.) Pakistan, during the growing season 2019. Peshawar is situated in Iranian plateau area having tropical climate. Soil texture was determined as sandy loam as evaluated via hydrometer method by [13].

The seeds of lentil (*Lens culinaris* Medik.) variety punjab-2009 were obtained from National Agriculture Research Centre (NARC) Islamabad, Pakistan. Surface sterilized seeds were sown 2–4 centimetres deep in the soil-filled earthen pots (20cm height, 18cm upper/lower diameter and 2cm thickness) with each pot containing 3kg sandy loam soil. Experiment was conducted in Randomized Complete Block Design (RCBD). Thinning and weeding were properly maintained and seedlings were exposed to sunlight for better growth. Three replicates were taken for each group. All the groups were normally watered till 60[th] day of emergence. Different levels (100, 200 and 300 mg/L) of α-tocopherol were prepared by mixing 100, 200 and 300 mg of α-tocopherol separately in 900 ml distilled water and 70% ethanol (9:1) followed by heating at 33˚C for 15 minutes. After 60 days of germination, one set of trial was exposed to 20-days of drought-induced stress and sprayed with three levels of α-tocopherol (100, 200, and 300 mg/L) once throughout the growing season. The second set of experiment was subjected to 25 days of drought-induced stress and sprayed with the same levels of exogenously applied α-tocopherol. At the end of drought-induced stress periods, five plants from each replicate were harvested randomly for the determination of various growth and physio-biochemical parameters.

### 2.2. Soil analysis

Over the last two years average soil chemical properties were as follows: electrical conductivity (EC) 2.67 ds/m, pH 6.3 [14], Nitrogen (N) content 4.02g/kg [15], organic Carbon (C) 23.3 g/kg [16], available potassium (K) 91.4 mg/kg [17] and Phosphorus (P) 8.1 mg/kg [18].

### 2.3. Growth measurements

Growth parameters: absolute growth rate (AGR), relative growth rate (RGR), coefficient of velocity of germination (CVG) and net assimilation rate (NAR) were calculated by following the formulas suggested by [19].

$$\text{AGR (plant height)} = \frac{H_2 - H_1}{t_2 - t_1} \qquad \text{(Eq 1)}$$

$H_1$ and $H_2$ denoted plant height (cm) during the time $t_1$ to $t_2$.

$$\text{RGR} = \frac{\log_e W_2 - \log_e W_1}{t_2 - t_1} \qquad \text{(Eq 2)}$$

$W_1$ and $W_2$ denoted plant dry weight (gm) at time $t_1$ and $t_2$, $\log_e$ is natural logarithm.

$$\text{CVG} = \frac{N_1 + N_2 + N_3 + \cdots + N_X}{100} (N_1 T_1 + N_2 T_2 + N_3 T_3 + \cdots + N_X T_X) \qquad \text{(Eq 3)}$$

The CVG indicates the pace of germination. Hypothetically, the maximum CVG possible is 100. This would happen if all seeds germinate on 1[st] day.

$$\text{NAR} = \frac{W_2 - W_1}{t_2 - t_1} \times \frac{\log_e A_2 - \log_1 A_1}{A_2 - A_1} \, (\text{g/cm/day}) \qquad \text{(Eq 4)}$$

$A_1$ and $A_2$ denoted surface area of leaf and $W_1$ and $W_2$ are plant total dry matter at Time $t_1$ and $t_2$.

Crop growth rate (CGR), leaf area index (LAI), leaf area ratio (LAR) and relative water content (RWC) were calculated using the following formulae suggested by [20]:

$$CGR = \frac{W2 - W1}{T2 - T1} \times \frac{1}{Land\ area}\ (g/m^2/d) \tag{Eq 5}$$

$W_1$ and $W_2$ are plant dry weights taken at time $T_1$ and $T_2$, respectively.

$$LAI = \frac{leaf\ area\ (cm)^2}{land\ area\ (cm)^2} \tag{Eq 6}$$

$$LAR = \frac{leaf\ area}{final\ plant\ dry\ weight} \tag{Eq 7}$$

$$RWC = \frac{Wf - Wd}{Ws - Wd} \times 100\ (\%) \tag{Eq 8}$$

"Wf" represented leaf fresh weight and "Wd" leaf dry weight. "Ws" indicated saturated weight of leaf material determined after floating the leaves in distilled water for 18 hours.

Root-shoot ratio (RSR) was calculated by using the formula proposed by [21]:

$$RSR = \frac{root\ dry\ mass}{shoot\ dry\ mass} \tag{Eq 9}$$

Seed vigor index (SVI) was measured by the formula proposed by [22]:

$$SVI = Seedling\ length\ (cm)\ x\ Seed\ germination\ \%\ age \tag{Eq 10}$$

## 2.4. Photosynthetic pigments (chlorophyll a, b & carotenoids)

Fresh leaf material (0.5 gm) was homogenized in 10 ml 80% acetone solution. Samples containing homogenized solution were kept in centrifuge machine and spun for 5 minutes. After centrifugation, the samples were kept in dark overnight at 4°C. On the following day, optical density of each sample was measured at 470, 645 and 663 nm for carotenoids and chlorophyll a & b quantification by following the protocol of [23].

## 2.5. Soluble sugar content (SSC)

Fresh foliar material (0.5 gm) was taken and grounded in 5 ml distilled water and homogenized mixture was prepared. The samples containing homogenized mixture were placed in centrifuge machine and spun for 10 minutes; after the process of centrifugation, 1ml supernatant was taken from each sample and 4 ml concentrated (35%) $H_2SO_4$ was added. Optical density (OD) was noted at 490 nm by adopting the methodology of [24].

## 2.6. Total proline content (TPC)

Proline content was quantified by the methodology of [25]. Fresh leaves (0.5 gm) were grounded in 10 ml 3% aqueous sulphosalicylic acid and a homogenized mixture was prepared. The mixture was filtered and 2 ml filtrate was taken. Similarly, 4 ml ninhydrin solution and 4 ml glacial acetic acid (20%) were mixed with 2 ml filtrate taken. The mixture was heated at 100°C for 1 hour and 4 ml toluene was added to it. OD readings were recorded at 520 nm.

## 2.7. Glycine betaine content (GBC)

Fresh foliar material (0.5gm) was chopped in 10 ml distilled water. The mixture was filtered; filtrate obtained was diluted by adding 2 ml $H_2SO_4$ solution. The samples were centrifuged for 10 minutes and Cold potassium iodide ($KI–I_2$) was added to supernatant. 1 ml supernatant was collected from each sample and optical density was measured at 365 nm by using the methodology of [26].

## 2.8. Soluble protein content (SPC)

Protein content in leaf tissues were investigated by following the protocol of [27]. 0.5 gm fresh leaf tissues were grounded in 5 ml phosphate buffer (pH 7.0) in ice cooled pestle and mortar. After grinding a homogenized mixture was obtained, samples from the prepared mixture were kept in centrifuge machine and spun for 15 minutes. After centrifugation, 0.1 ml supernatant was taken from each sample and 2 ml Bradford reagent was added. Optical density was measured at 595 nm.

## 2.9. Malondialdehyde content (MDAC)

Fresh leaf material (0.25 gm) was chopped in 3 ml 1.0% (w/v) Trichloro acetic acid (TCA) and mixture was prepared. Samples containing homogenized mixture were kept in centrifuge machine and spun in for 10 minutes. After the process of centrifugation, 1 ml supernatant was taken and 4 ml 0.5% (w/v) 2-thiobarbituric acid was added. Samples were heated at 95˚C for 1 hour and then cooled by placing in ice bath for 10 minutes. Optical density was measured at 532 nm by the following the method of [28].

## 2.10. Hydrogen peroxide content (HPOC)

Fresh foliar material (0.5 gm) was chopped in 5 ml trichloro acetic acid (TCA) and homogenous mixture was prepared. The samples were placed in centrifuge machine and rotated for 15 minutes. After centrifugation, 0.5 ml supernatant was taken from each sample and 0.5 ml phosphate buffer and 1.0 ml potassium iodide (KI) reagent was added to it. Optical density was recorded at 390 nm by following the methodology of [29].

## 2.11. Endogenous tocopherol content (ETPC)

Methodology of [30] was followed to measure the levels of endogenous tocopherol content in leaf tissues. Leaf material (0.1 g) was grounded in 10 ml solution (petroleum ether and ethanol 2:1.6 v/v) and homogenized mixture was prepared. Samples containing homogenized mixture were placed in centrifuge and spun for 15 minutes. After centrifugation, 1 ml supernatant was taken and mixed with 0.2 ml (2%) 2-dipyridyl in ethanol (v/v). The mixture was poured into cuvete and placed in spectrophotometer. Optical density was measured at 520 nm.

## 2.12. Ascorbate peroxidase activity (APX)

Ascorbate peroxidase (APX) levels were evaluated by pursuing the method of [31]. Fresh leaf material (0.5 gm) was grounded in 5 ml phosphate buffer (pH 7.0). The samples were placed in centrifuge machine and rotated for 15 minutes. After the process of centrifugation, 0.2 ml supernatant was collected from each sample and 0.1 mM hydrogen peroxide, 0.6 mM ascorbic acid and 0.1 mM ethylenediamine tetraacetic acid (EDTA) was added. Optical density was recorded at 290 nm.

### 2.13. Catalase activity (CAT)

Catalase activity was measured by following the protocol of [32]. Fresh foliar material (0.5gm) was homogenized in 5 ml buffer solution (pH 7.0). The samples of the mixture were kept in centrifuge machine and rotated at 3000 rpm for 15 minutes. After centrifugation, 0.1 ml supernatant was taken and 1.9 ml phosphate buffer (50 Mm) and 0.1 ml $H_2O_2$ (5.9 mM) was added. Optical density readings were noted at 240 nm for 3 minutes.

### 2.14. Superoxide dismutase activity (SOD)

Fresh foliar material (0.5 gm) was grounded in 5 ml phosphate buffer and homogenized mixture was prepared. The samples of the mixture were placed in centrifuge machine and spun for 15 minutes. After centrifugation, 0.1 ml supernatant was taken from each sample and 5 ml methionine, 150 µl riboflavin and 24 µl nitro-blue-tetrazolium (NBT) were added. Optical density was recorded at 560 nm by pursuing the method of [33].

### 2.15. Peroxidase activity (POD)

Peroxidase (POD) activity was determined by following the methodology of [32]. Leaf material (0.5 gm) was chopped in 2 ml morpholino ethane sulphonic acid (MES) and homogenized mixture was prepared. The samples were placed in centrifuge machine and spun for 15 minutes. After centrifugation, 0.1 ml supernatant was collected from each sample and 1.3 ml MES, 0.1 ml phenyl diamine and 1 ml hydrogen peroxide (30%) were added. OD was noted at 470 nm for 3 minutes via spectrophotometer.

### 2.16. Statistical analysis

The experiment comprised of two factors including drought induced stress of 20 and 25 days and α-tocopherol levels, 100, 200 and 300 mg/L. Randomized complete block design (RCBD) was adopted for experiment and three replicates were taken for each group. SPSS Statistic-25 software was used for analysis of variance (ANOVA). By using standard techniques, mean and standard errors were calculated and least significance difference (LSD) test at ($p \leq 0.05$) was performed and indicated by letters (A-E). Correlation analysis was performed by using Statistix 8.1 software.

## 3. Results

### 3.1. Growth responses under drought induced stress and α-tocopherol levels

Statistical analysis revealed a significant increase at ($P \leq 0.05$) in AGR, CGR, RGR and LAI with exogenously applied α-tocopherol 200 mg/L in comparison with control group and rest of the treatments under 20-days of drought induced stress. On the contrary, these parameters were affected negatively under 20-days of drought induced stress with no α-tocopherol treatment (Fig 1A–1D). NAR and CVG showed improvement at ($P \leq 0.05$) in control group and group with 100 mg/L α-tocopherol treatment (Figs 2A and 3B). Furthermore, RSR showed significant improvement at ($P \leq 0.05$) with α-tocopherol 200 mg/L and was affected adversely on exposure to drought induced stress with no α-tocopherol application (Fig 2C). In contrast with other treatments, RWC was recorded maximum in the control group only (Fig 2D). In comparison with the rest of the treatments and control group LAR showed significant results at ($P \leq 0.05$) with 200 mg/L α-tocopherol. On the contrary, LAR was adversely affected under induced water stress condition with no α-tocopherol treatment. SVI showed positive response

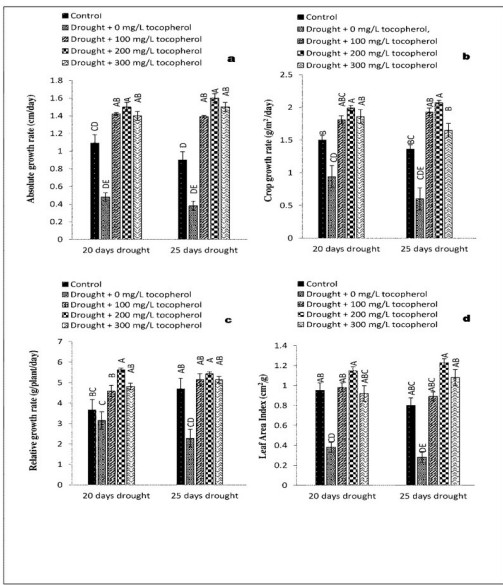

**Fig 1.** Effect of varying levels of exogenously applied α-tocopherol on absolute growth rate (**a**) crop growth rate (**b**) relative growth rate (**c**) leaf area index (**d**) of lentil (*Lens culinaris* Medik.) grown under varying drought stress condition (Mean ± standard error.) letters (A–E) indicating least significance difference among the mean values at $p \leq 0.05$.

at ($P \leq 0.05$) in control group only (Fig 3A and 3B). Same growth parameters were studied under 25-days of drought induced stress sprayed with the same levels of α-tocopherol. Results showed significant improvement at ($P \leq 0.05$) in AGR, LAI, and LAR with the application of

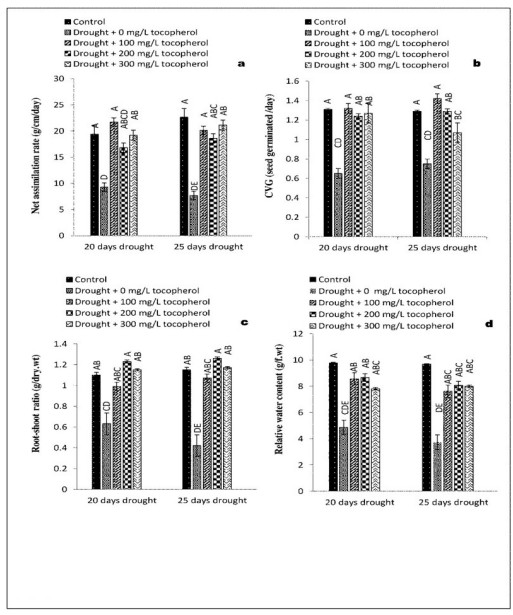

**Fig 2.** Effect of varying levels of exogenously applied α-tocopherol on net assimilation rate (a) coefficient of velocity of germination (b) root-shoot ratio (c) relative water content (d) of lentil (*Lens culinaris* Medik.) grown under varying drought stress condition (Mean ± standard error.) letters (A–E) indicating least significance difference among the mean values at $p \leq 0.05$.

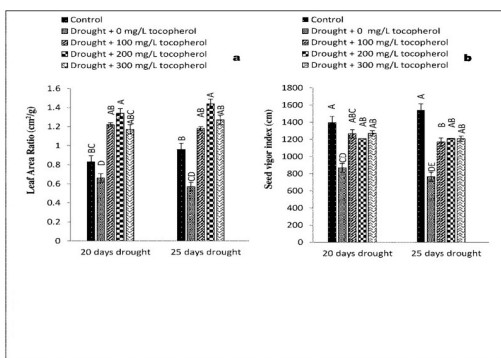

**Fig 3.** Effect of varying levels of exogenously applied α-tocopherol on leaf area ratio (**a**) seed vigor index (**b**) of lentil (*Lens culinaris* Medik.) grown under varying drought stress condition (Mean ± standard error.) letters (A–E) indicating least significance difference among the mean values at $p \leq 0.05$.

200 mg/L α-tocopherol only. On the other hand, AGR, CGR, LAI, LAR and NAR showed negative responses under drought induced stress of 25-days with no α-tocopherol application. Highest NAR and CVG values were calculated for control group and 100 mg/L α-tocopherol treatment in contrast with the rest of the treatments.

RGR and RSR both showed significant enhancement at ($P \leq 0.05$) with the application of 200 mg/L α-tocopherol; however, both the parameters were affected negatively on exposure to 25-days of drought induced stress with no α-tocopherol treatments. Subsequently, in comparison with the rest of treatments RWC and SVI values were highest only in the control group.

## 3.2. Determination of photosynthetic pigments (chlorophyll a, b & carotenoids)

Varying levels of drought induced stress markedly ($P \leq 0.05$) reduced chlorophyll "a" content. Application of α-tocopherol enhanced ($P \leq 0.01$) chlorophyll "a" content in both the drought induced stress levels. In contrast with control group and rest of the treatments sprayed with 200 mg/L tocopherol showed maximum chlorophyll "a" content (Table 1 and Fig 4A). Drought stress condition significantly (P≤0.01) decreased chlorophyll "b" content; however, α-tocopherol ameliorated ($P \leq 0.05$) the levels of chlorophyll "b" content. Among all the treatments and control group application of α-tocopherol 200 mg/L showed better response in improving the levels of chlorophyll "b" content (Table 1 and Fig 4B).

Drought induced stress regimes considerably ($P \leq 0.001$) reduced the levels of total chlorophyll content, among all the treatments and control group. α-tocopherol 200 mg/L boosted ($P \leq 0.001$) the levels of total chlorophyll content in both (20 and 25 days) of drought induced stress levels (Table 1 and Fig 4C). Drought stress condition reduced ($P \leq 0.001$) the levels of carotenoid content to a great extent. As compared to other treatments and control, 100 mg/L of α-tocopherol application showed significant ($P \leq 0.001$) response in increasing the levels of carotenoid content (Table 1 and Fig 4D).

## 3.3. Changes in concentration of soluble sugar content (SSC)

Drought stress condition had a significant influence on the concentration of soluble sugar content. Limited water regimes raised ($P \leq 0.001$) the levels of soluble sugar content in both the drought levels. Exogenously applied α-tocopherol further raised ($P \leq 0.001$) the levels of soluble sugar content. In comparison with control and other applied treatments, α-tocopherol 200 mg/L proved more effective in ameliorating soluble sugar content (Table 1 and Fig 5A).

**Table 1. Analysis of variance of physio-biochemical attributes of lentil cultivar to varying levels of α-tocopherol under drought induced stress.**

| Traits | Source of variation | SS | Df | MS | F | P |
|---|---|---|---|---|---|---|
| Chl "a" | Cultivar | 0.162 | 2 | 0.081 | 0.664 | 0.523 |
| | Drought | 2.023 | 9 | 0.225 | 3.135 | 0.016* |
| | Drought × Treatment | 1.861 | 7 | 0.266 | 3.708 | 0.001** |
| | Error | 1.434 | 20 | 0.072 | - | - |
| Chl "b" | Cultivar | 0.557 | 2 | 0.278 | 2.993 | 0.067* |
| | Drought | 1.936 | 9 | 0.215 | 3.802 | 0.006** |
| | Drought × Treatment | 1.379 | 7 | 0.197 | 3.482 | 0.013* |
| | Error | 1.131 | 20 | 0.057 | - | - |
| TCC | Cultivar | 1.226 | 2 | 0.613 | 2.043 | 0.149 |
| | Drought | 7.316 | 9 | 0.813 | 8.091 | 0.000*** |
| | Drought × Treatment | 6.090 | 7 | 0.870 | 8.659 | 0.000*** |
| | Error | 2.009 | 20 | 0.100 | - | - |
| CC | Cultivar | 0.771 | 2 | 0.385 | 1.134 | 0.337 |
| | Drought | 7.765 | 9 | 0.863 | 7.900 | 0.000*** |
| | Drought × Treatment | 6.995 | 7 | 0.999 | 9.148 | 0.000*** |
| | Error | 2.184 | 20 | 0.109 | - | - |
| SSC | Cultivar | 8.585 | 2 | 4.292 | 40.732 | 0.000*** |
| | Drought | 10.392 | 9 | 1.155 | 22.250 | 0.001*** |
| | Drought × Treatment | 1.807 | 7 | 0.258 | 4.975 | 0.002*** |
| | Error | 1.038 | 20 | 0.052 | - | - |
| SPC | Cultivar | 0.676 | 2 | 0.338 | 0.961 | 0.395 |
| | Drought | 7.843 | 9 | 0.871 | 7.478 | 0.000*** |
| | Drought × Treatment | 7.167 | 7 | 1.024 | 8.785 | 0.000*** |
| | Error | 2.331 | 20 | 0.117 | | |
| TPC | Cultivar | 8.769 | 2 | 4.385 | 70.578 | 0.000*** |
| | Drought | 9.749 | 9 | 1.083 | 31.064 | 0.000*** |
| | Drought × Treatment | 0.980 | 7 | 0.140 | 4.014 | 0.007** |
| | Error | 0.697 | 20 | 0.035 | - | - |
| POD | Cultivar | 4.433 | 2 | 2.216 | 28.347 | 0.005** |
| | Drought | 5.315 | 9 | 0.591 | 9.611 | 0.002** |
| | Drought × Treatment | 0.882 | 7 | 0.126 | 2.051 | 0.008** |
| | Error | 1.229 | 20 | 0.061 | - | - |
| SOD | Cultivar | 8.353 | 2 | 4.177 | 36.118 | 0.002** |
| | Drought | 9.876 | 9 | 1.097 | 13.720 | 0.004** |
| | Drought × Treatment | 1.523 | 7 | 0.218 | 2.720 | 0.007** |
| | Error | 1.600 | 20 | 0.080 | - | - |
| APX | Cultivar | 8.394 | 2 | 4.197 | 16.699 | 0.034* |
| | Drought | 10.032 | 9 | 1.115 | 11.512 | 0.002** |
| | Drought × Treatment | 1.638 | 7 | 0.234 | 2.417 | 0.006** |
| | Error | 1.936 | 20 | 0.097 | - | - |
| CAT | Cultivar | 9.864 | 2 | 4.932 | 59.922 | 0.005 |
| | Drought | 11.200 | 9 | 1.244 | 12.849 | 0.003** |
| | Drought × Treatment | 1.336 | 7 | 0.191 | 10.866 | 0.001** |
| | Error | 0.351 | 20 | 0.018 | - | - |
| HPOC | Cultivar | 1.122 | 2 | 0.561 | 2.070 | 0.146 |
| | Drought | 4.910 | 9 | 0.546 | 3.091 | 0.017* |
| | Drought × Treatment | 3.788 | 7 | 0.541 | 3.065 | 0.023* |

*(Continued)*

**Table 1.** (Continued)

| Traits | Source of variation | SS | Df | MS | F | P |
|---|---|---|---|---|---|---|
| | Error | 3.531 | 20 | 0.177 | - | - |
| ETPC | Cultivar | 4.227 | 2 | 2.113 | 15.269 | 0.000*** |
| | Drought | 5.709 | 9 | 0.634 | 5.629 | 0.001** |
| | Drought × Treatment | 1.483 | 7 | 0.212 | 1.880 | 0.007** |
| | Error | 2.254 | 20 | 0.113 | - | - |
| MDAC | Cultivar | 1.035 | 2 | 0.517 | 3.927 | 0.032* |
| | Drought | 3.820 | 9 | 0.424 | 11.020 | 0.000*** |
| | Drought × Treatment | 2.786 | 7 | 0.398 | 10.331 | 0.000*** |
| | Error | 0.770 | 20 | 0.039 | - | - |
| GBC | Cultivar | 3.628 | 2 | 1.814 | 13.427 | 0.000*** |
| | Drought | 3.804 | 9 | 0.423 | 2.435 | 0.047* |
| | Drought × Treatment | 0.176 | 7 | 0.025 | 0.145 | 0.003** |
| | Error | 3.472 | 20 | 0.174 | - | - |

Chl "a" = Chlorophyll "a", Chl "b" = Chlorophyll "b", TCC = Total chlorophyll content, CC = Carotenoid content, SSC = Soluble sugar content, SPC = Soluble protein content, TPC = Total proline content, POD = Peroxidase, SOD = Superoxide dismutase, APX = Ascorbate peroxidase, CAT = catalase, HPOC = Hydrogen peroxide content, ETPC = Endogenous tocopherol content, MDAC = Malondialdehyde content, GBC = Glycine betaine content. (* Significant at the $P = 0.05$, ** Significant at the $P = 0.01$, *** Significant at the $P = 0.001$) SS = sum of square, Df = degree of freedom, MS = mean square, F = variation between sample means, $P$ = probability value.

### 3.4. Effects on soluble protein content (SPC)

Drought stress inflicted a profound effect on soluble protein content as it markedly reduced ($P \leq 0.001$) the levels of soluble protein content. Among all the treatments, α-tocopherol 200 mg/L was significant in raising the levels of soluble protein content (Table 1 and Fig 5B).

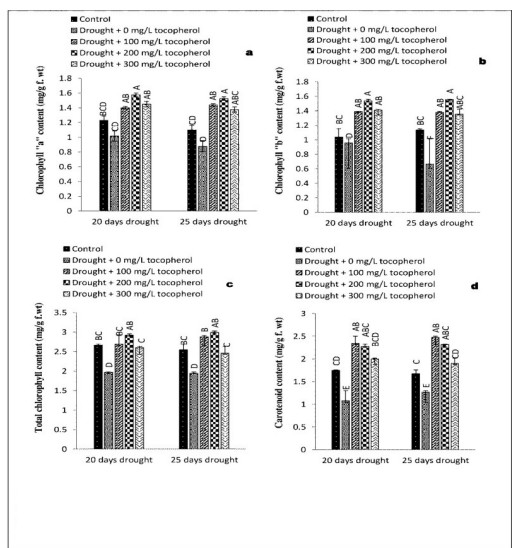

**Fig 4.** Effect of varying levels of exogenously applied α-tocopherol on chlorophyll "a" content (**a**) chlorophyll "b" content (**b**) total chlorophyll content (**c**) carotenoid content (**d**) of lentil (*Lens culinaris* Medik.) grown under varying drought stress condition (Mean ± standard error.) letters (A–F) indicating least significance difference among the mean values at $p \leq 0.05$.

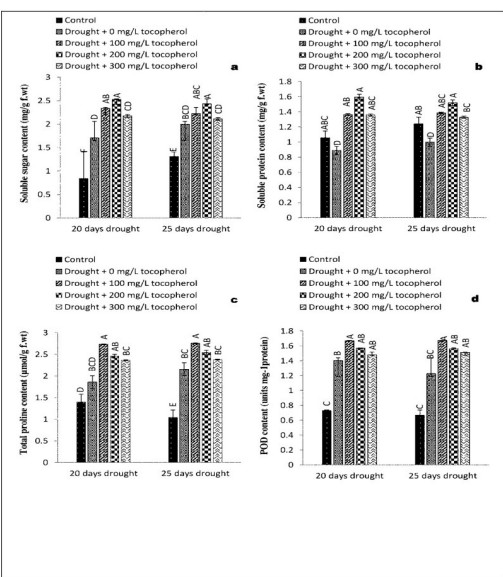

**Fig 5.** Effect of varying levels of exogenously applied α-tocopherol on soluble sugar content (**a**) soluble protein content (**b**) total proline content (**c**) peroxidase content (**d**) of lentil (*Lens culinaris* Medik.) grown under varying drought stress condition (Mean ± standard error.) letters (A–F) indicating least significance difference among the mean values at $p \leq 0.05$.

### 3.5. Fluctuations in the levels of total proline content (TPC) and Glycine betaine content (GBC)

Drought stress condition increased total soluble proline content to a considerable level ($P \leq 0.001$). Application of α-tocopherol further enhanced ($P \leq 0.01$) soluble proline content under varying drought stress regimes. Among different α-tocopherol levels, 100 mg/L showed better results (Table 1 and Fig 5C).

### 3.6. Responses of antioxidant enzymes activities (POD, SOD, APX & CAT)

On exposure to drought induced stressed condition a significant increase was noted in the activities of POD ($P \leq 0.01$) (Table 1 and Fig 5D), SOD ($P \leq 0.01$), APX ($P \leq 0.01$), and CAT ($P \leq 0.01$). Foliar application of α-tocopherol further significantly ($P \leq 0.01$) enhanced the activities of these enzymes. In case of peroxidase and superoxide dismutase 100 mg/L α-tocopherol treatment proved more effective while in the case of ascorbate peroxidase and catalase 200 mg/L α-tocopherol responded better as compared to the rest of the treatments and control group (Table 1 and Fig 6A–6C).

### 3.7. Changes in the concentration of hydrogen peroxide content ($H_2O_2$)

Drought stress condition caused a marked ($P \leq 0.05$) increase in hydrogen peroxide concentration. In comparison with control group and rest of the applied treatments α-tocopherol 200 and 300 mg/L showed a significant ($P \leq 0.05$) response in alleviating the levels of hydrogen peroxide content under limited water condition (Table 1 and Fig 6D).

### 3.8. Levels of endogenous tocopherol content (ETPC)

Drought stress condition resulted in the accumulation of endogenous/membrane bounded tocopherol content to a level significant at ($P \leq 0.01$). Among all the foliar applied α-tocopherol

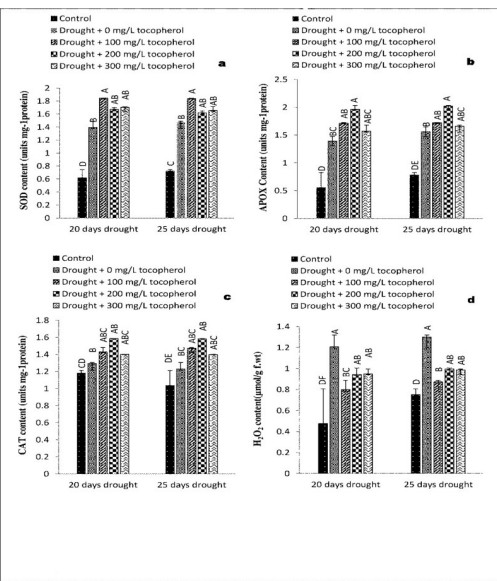

**Fig 6.** Effect of varying levels of exogenously applied α-tocopherol on superoxide dismutase content (**a**) ascorbate peroxidase content (**b**) catalase content (**c**) hydrogen peroxide content (**d**) of lentil (*Lens culinaris* Medik.) grown under varying drought stress condition (Mean ± standard error.) letters (A–E) indicating least significance difference among the mean values at $p \leq 0.05$.

treatments 100 mg/L tocopherol increased the levels of endogenous tocopherol content to a significant ($P \leq 0.01$) level (Table 1 and Fig 7A).

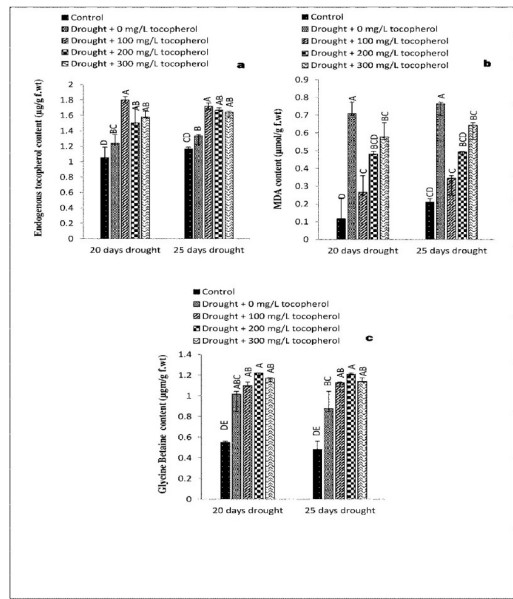

**Fig 7.** Effect of varying levels of exogenously applied α-tocopherol on endogenous tocopherol content (**a**) malondialdehyde content (**b**) glycine betaine content (**c**) of lentil (*Lens culinaris* Medik.) grown under varying drought stress condition (Mean ± standard error.) letters (A–E) indicating least significance difference among the mean values at $p \leq 0.05$.

## 3.9. Changes in the concentration of Malondialdehyde (MDAC) content

Water deficit stress significantly (P≤0.001) raised the concentration of malondialdehyde content. On the contrary, α-tocopherol application reduced the concentration of malondialdehyde to a considerable level (*P*≤0.001). However, in both the drought levels α-tocopherol 100 mg/L showed better response in decreasing the concentrations of malondialdehyde (Table 1 and Fig 7B).

## 3.10. Fluctuations in the levels of total Glycine betaine content (GBC)

Water deficiency triggered a marked (*P*≤0.05) increase in glycine betaine content. Foliar applied α-tocopherol further improved the concentration of glycine betaine content. α-tocopherol 200 mg/L showed better response in increasing the concentrations of glycine betaine content (Table 1 and Fig 7C).

## 3.11. Correlation between drought induced stress, α-tocopherol and physio-biochemical attributes

Positively significant (*P*≤0.05) correlation was noted between varying drought stressed condition and physio-biochemical attributes of lentil including malondialdehyde, hydrogen peroxide, glycine betaine, soluble sugar, total proline, endogenous tocopherol contents and activities of antioxidant enzymes including peroxidase, superoxide dismutase, catalase and ascorbate peroxidase; whereas, chlorophyll a, b, total chlorophyll content, carotenoid content, and total soluble protein content correlated negatively with both 20 and 25 days of drought induced stressed regimes. A positive and significant correlation was observed between varying levels of exogenously applied α-tocopherol and chlorophyll a, b, total chlorophyll content, carotenoid content, soluble sugar content, soluble protein content, total proline content, endogenous tocopherol content, glycine betaine content and activities of antioxidant enzymes (APX, CAT, SOD and POD); however, a significantly negative correlation was observed between α-tocopherol levels and concentrations of hydrogen peroxide and Malondialdehyde content (Table 2).

# 4. Discussion

## 4.1. Growth responses

Oxidative stress and higher production of ROS have been considered the major causes that adversely affect plant growth [34]. Reduction in plant growth parameters such as LAI and LAR have been reported to be mainly caused by stomatal closure during drought stress condition,

**Table 2. Correlation between induced drought stress, α-tocopherol and physio-biochemical attributes of lentil cultivar.**

| Correlation analysis between drought stress and physio-biochemical attributes | | | | | | | | | | | | | | |
|---|---|---|---|---|---|---|---|---|---|---|---|---|---|---|
| Chl "a" | Chl "b" | TCC | CC | SSC | SPC | TPC | POD | SOD | APX | CAT | HPOC | ETPC | MDAC | GBC |
| -0.421* | -0.562* | -0.601* | -0.654* | 0.862* | -0.854* | 0.907* | 0.799* | 0.885* | 0.854* | 0.833* | 0.968* | 0.777* | 0.961* | 0.882* |
| **Correlation analysis between tocopherol and physio-biochemical attributes** | | | | | | | | | | | | | | |
| Chl "a" | Chl "b" | TCC | CC | SSC | SPC | TPC | POD | SOD | APX | CAT | HPOC | ETPC | MDAC | GBC |
| 0.893* | 0.884* | 0.899* | 0.903* | 0.911* | 0.941* | 0.993* | 0.887* | 0.918* | 0.889* | 0.924 | -0.551* | 0.889* | -0.647* | 0.919* |

Chl "a" = Chlorophyll "a", Chl "b" = Chlorophyll "b", TCC = Total chlorophyll content, CC = Carotenoid content, SSC = Soluble sugar content, SPC = Soluble protein content, TPC = Total proline content, POD = Peroxidase, SOD = Superoxide dismutase, APX = Ascorbate peroxidase, CAT = catalase, HPOC = Hydrogen peroxide content, ETPC = Endogenous tocopherol content, MDAC = Malondialdehyde content, GBC = Glycine betaine content. (* Significant at the *P* = 0.05, ** Significant at the *P* = 0.01, *** Significant at the *P* = 0.001).

that eventually leads to leaf senescence [35]. On exposure to drought stress condition, plants tend to close their stomata which slow down the water loss from aerial parts of the plants. As a result, the carbon dioxide ($CO_2$) absorption is reduced and consequently net photosynthesis [36]. Moreover, reduction in growth attributes such as CVG, CGR, AGR, RGR, RSR, SVI and NAR are closely linked with suppression of cell elongation and cell growth under drought stress regimes [37]. In the present study, induced water deficit stress impacted negatively on lentil growth and physio-biochemical mechanisms. On the contrary, foliar applied α-tocopherol curbed the negative consequences of drought induced stress by regulating key metabolic activities and boosted the growth parameters including AGR, CGR, LAI, LAR, NAR and CVG (Figs 1–3). Induced water deficit stress caused a considerable reduction in AGR, CGR, LAI, LAR and NAR of lentil. Likewise, [38,39] also observed reduction in these growth parameters under drought induced stress conditions. It is believed that this reduction is due to stomatal closure, persistent exposure to limited water regimes ultimately leading to shrinkage of leaves [38,40]. Similar variations in growth parameters were recorded by other workers [41,42]. Results from statistical analysis in Figs 1–3 indicated that RSR, RGR, RWC and SVI were improved in the control group and affected negatively in groups subjected to varying drought stress levels with no α-tocophrol treatments, confirming the same investigations made by [43].

## 4.2. Physiological and biochemical responses

Under drought stress conditions, an increase in chlorophyll degrading enzymes activity prompts the destruction of chlorophyll pigments [44]. Drought stress causes chlorophyll pigments degeneration and hampering the process of photosynthesis [45,46]. Chloroplast is the most sensitive organelle to drought stress exposure, and it has been proved that drought stress damages photosynthetic pigments in various crops [47,48].

It is suggested that the degeneration of chlorophyll is associated with the production of ROS, which lowers down the rate of photosynthesis and increases cellular respiration [49,50]. In the present study, photosynthetic pigments (chlorophyll a, b and carotenoid content) were affected negatively on exposure to drought induced stress. In contrast, exogenously applied α-tocopherol significantly enhanced the levels of photosynthetic pigments (Fig 4A–4D). α-tocopherol protects the chloroplast membranes from photo oxidation and assists to provide suitable environment for the photosynthetic machinery to work efficiently under oxidative stress condition. Accumulation of tocopherol in biological membranes occurs as a response to various types of abiotic stresses including drought, high temperature, salinity and cold [51]. Additionally, tocopherol contributes to membrane integrity and stability by manipulating its permeability and fluidity [52]. Our results were in accordance with the previous research work carried out by [10,53,54].

In response to abiotic stress, plants perceive a disturbance in their physiological activities and respond abruptly by accumulating a variety of osmolytes mainly including glycine betaine and proline. Osmolytes provide suitable environment for various metabolic activities and protecting plants from the damages caused by oxidative stress described by [55]. Compounds like proline, MDA and $H_2O_2$ are generally being used as stress markers. Proline is well known for its osmoprotective role. In many plants an increased concentration of proline during drought stress condition indicated to be correlated with drought stress tolerance [56]. It is also evident that proline has the potential to directly act as ROS scavenger and regulator of cellular redox status [57]. The studied lentil cultivar revealed a marked rise in soluble sugar content (Fig 5A), total proline content (Fig 5C), and glycine betaine content (Fig 7C) by experiencing water deficit stress condition. Similarly, foliar applied α-tocopherol further increased the concentrations of these osmolytes. Our results were consistent with the investigation made by [58] in Chinese

rye grass (*Leymus chinensis*) in which a high level of proline was observed in drought stressed seedlings grown from seeds primed with α-tocopherol. The same results were recorded by [59] in soybean (*Glycine max*) and in faba bean (*Vicia faba*) by [60].

Water-limited condition causes protein and lipid degeneration and affect plant growth and other vital activities [61]. In case of lentil, it was found that low moisture content in the soil alleviated the soluble protein content to a great extent. Application of α-tocopherol (200 mg/ L) was significant in raising the levels of soluble protein content (Fig 5B). [54] Concluded that flax (*Linum usitatissimum*) plants sprayed with α-tocopherol under salinity stress caused a marked increase in soluble protein content. Similarly, [60] suggested that application of (100 mg/L) of α-tocopherol in faba bean (*Vicia faba*) plants proved significant in preservation of soluble protein content.

The main role of *a*-tocopherol is the removal of lipid peroxyl radical prior of its attack to target lipid substrate synthesizing *a*-tocopheroxyloxyl radicals [62]. Under abiotic stress condition, α-tocopherol deactivates $^1O_2$ in chloroplast; according to an estimate a single molecule of α-tocopherol can deactivate 120 molecules of $^1O_2$ [63]. Membrane bounded tocopherol levels were observed maximum in lentil plants exposed to drought stress regimes. Application of α-tocopherol (100 mg/L) proved more significant in raising the levels of membrane bounded tocopherol content (Fig 7A). Our findings in the case of lentil were supported by the investigations made by [64] who reported high levels of endogenous tocopherol contents in maize plants cultivated under induced water stress condition [65]. Confirmed same results in canna (*Canna edulis*) cultivars under induced drought stress condition.

Malondialdehyde (MDA) is the product of membrane degradation. Under abiotic stress condition, a rise in MDA concentration marks the disintegration of biological membranes [66]. Accumulation of MDA has been considered an indication of lipid peroxidation in various plants under stress condition [67,68]. In plant tissues peroxidation of free fatty acid could occur both in non-enzymatic and enzymatic ways, producing a number of breakdown products which mainly include alcohol, aldehydes and their esters and this process is considered to be mainly involved in oxidative damage to cellular membranes and other biomolecules [69]. Physiological analysis of lentil cultivar revealed an increase in MDA content under drought stress condition. Whereas, exogenously applied α-tocopherol significantly decreased levels of MDA content (Fig 7B). The same results were obtained by [70] in case of geranium (*Pelargonium graveolens*) treated with (100 mg/L) α-tocopherol.

Hydrogen peroxide ($H_2O_2$) being a prominent reactive oxygen species (ROS) in plants results in cell oxidation, disturbs vital metabolic processes and interrupts membrane stability under stress condition [71]. Plants naturally produce ROS, mainly including $H_2O_2$ superoxides, there is a delicate balance between ROS production and it's scavenging, under drought stress condition this balance is disturbed as plant tend to close their stomata which also limits $CO_2$ fixation [72]. In present study, a significant increase was observed in $H_2O_2$ content under drought stress condition. Application of α-tocopherol decreased the concentration of $H_2O_2$ to a considerable level (Fig 6D). Parallel to our results, [73] recorded that application of resveratrol and α-tocopherol decreased the levels of $H_2O_2$ in citrus plants subjected to salinity stress. The same results were reported by [74] who found that (100 mg/L) α-tocopherol proved better in lowering the levels of $H_2O_2$ in wheat plants exposed to salinity stress.

Plants counteract the oxidative stress generated by ROS in a coordinated way both enzymatically and non-enzymatically [75,76]. SOD act as first line of defence as it converts the superoxide radicals to $H_2O_2$ [46,77]. Likewise, [78] in their demonstration on *Carthamus tinctorius* cultivars subjected to drought stress, concluded that enzymatic as well as non-enzymatic antioxidants were involved in the removal of ROS, SOD is needed to scavenge superoxide radical [79], while scavenging $H_2O_2$, requires POD, CAT and APX [80]. Natural self-defence

systems are well developed in plants. On exposure to stress condition these defence systems are activated both enzymatically (ascorbate peroxidase, catalase, superoxide dismutase and peroxidase) and non-enzymatically (secondary metabolites) scavenging the reactive oxygen species (ROS) formed as result of stress condition [76]. Though, activities and response of these antioxidants varies plant to plant [78]. In the present research study, water stress condition caused a prominent increase in the activities of antioxidant enzymes such as, POD (Fig 5D), SOD, APX, and CAT (Fig 6A–6C). However, foliar applied α-tocopherol further enhanced the activities of these enzymes. Our results are in agreement with the findings made by [58] who observed a marked increase in SOD and POD activities in Chinese rye grass (*Leymus chinensis*) subjected to induced water stress condition. Similarly, [81] reported an increase in the activities of POD and CAT in sunflower plants under salt stress. Furthermore, [60] reported that, application of α-tocopherol (200 mg/L) enhanced the performance of antioxidant enzymes to a great extent in drought stress plants of faba beans (*Vicia faba*).

## 5. Conclusions

From the present study, it was concluded that application of exogenous α-tocopherol under drought induced stress conditions prevented membrane degeneration by hampering lipid peroxidation, improving the levels of osmolytes such as glycine betaine, proline, protein and sugar. Moreover, exogenously applied α-tocopherol ameliorated the activities of antioxidant enzymes including APX, CAT, POD, and SOD. Resultantly, regulating various physio-biochemical and growth attributes. Furthermore, application of α-tocopherol (200 mg/L) followed by (100 mg/L) showed better responses in mitigating the damaging effects of oxidative stress. Importantly, the present scenario of alarmingly increasing changes in climatic conditions calls for a dire need of further research studies to investigate various growth and physiological responses of lentil cultivars to different types of biotic and abiotic stresses.

## Acknowledgments

We are highly acknowledged to Department of Botany, University of Peshawar for providing all facilities regarding this work.

## Author Contributions

**Investigation:** Sajjad Ali, Muhammad Idrees, Muhammad Nauman Khan.

**Software:** Kashif Ali, Ajmal Khan.

**Supervision:** Sami Ullah.

**Writing – original draft:** Wadood Shah.

**Writing – review & editing:** Muhammad Ali, Farhan Younas.

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
