## [Decision Letter · Decision Letter 0]

22 Mar 2021

PONE-D-21-05714

Effect of exogenously applied alpha-tocopherol on vital agronomic, physiological and biochemical attributes of Lentil (Lens culinaris Medik.) under induced drought stress

PLOS ONE

Dear Dr. Sami Ullah,

Thank you for submitting your manuscript to PLOS ONE. After careful consideration, we feel that it has merit but does not fully meet PLOS ONE’s publication criteria as it currently stands. Therefore, we invite you to submit a revised version of the manuscript that addresses the points raised during the review process.

I agree with the reviewers that language editing is needed. Abstract need to be modified. Correction should be done in materials and methods according to reviewers. Quality of graphs need to improve. The discussion of the results must be done in scientific terms, explaining the meaning of the results. Conclusion should be improved.

We look forward to receiving your revised manuscript.

Kind regards,

Basharat Ali, Ph.D

Academic Editor

PLOS ONE

Journal Requirements:

Reviewers' comments:

Reviewer's Responses to Questions

**Comments to the Author**

1. Is the manuscript technically sound, and do the data support the conclusions?

Reviewer #1: Yes

Reviewer #2: Partly

2. Has the statistical analysis been performed appropriately and rigorously? 

Reviewer #1: Yes

Reviewer #2: Yes

3. Have the authors made all data underlying the findings in their manuscript fully available?

Reviewer #1: Yes

Reviewer #2: Yes

4. Is the manuscript presented in an intelligible fashion and written in standard English?

Reviewer #1: Yes

Reviewer #2: No

5. Review Comments to the Author

Reviewer #1: The authors have presented the effects of drought induced stress by exogenous application of alpha-tocopherol on the agronomic and physiological attributes of lentil. The manuscript shows mechanistic approach and provides extensive results that could be useful in scientific community. However, this manuscript could not be considered for publication in its present form. Some comments are available for revision in order to improve the manuscript:

Major comments:

1. Highlights are missing. Add them substantially.

Minor comments:

Title should be revised. Either change the title by giving the punch line of your findings as a title or keep the same in this form “Effect of exogenously applied alpha-tocopherol on vital agronomic, physiological and biochemical attributes of Lentil (Lens culinaris Medik.) under drought induced stress”

Line 31: Revise it to “climatic conditions”

Line 32: “ill” must be replaced with “drastic”

Line 33: Revise “induced drought” to “drought induced”

Line 34: Mention the abbreviation of “Alpha α” at the first place (line 32) and then use it throughout.

Line 66: Add Comma after “findings”

Line 74: Insert Comma after “plants”

Line 79: Insert Comma after “regimes”

Line 83: Full stop after “Fabaceae”

Line 88: Change “induced drought stress” to “drought induced stress” in the whole manuscript.

Line 113: Revise “potassium available” to “available potassium”

Line 123: Please provide some detail about CVG in materials and methods.

Line 204: Seems like Alpha-tocopherol levels??

Line 205: Inset TAB at the start of the paragraph.

Line 211: Insert comma after “treatments”

Line 227: Add Full stop after “error”. Follow this for all the figures.

Line 239: Delete the word “group” after treatments. You have used this repeatedly. Please revise it carefully throughout as it is not making a good sentence structure.

Line 250: It is better to mention all the figure legends together after the references. Please revise it. Follow this for all the legends.

Line 262: Add comma after treatments.

Line 266: Delete the word “condition”

Line 338: Add Comma after “condition”

Line 338: “slow” must be replaced with “slows”

Line 342: Add comma after “study”

Line 343: Add comma after “contrary”

Line 350: Remove parenthesis after “in”

Line 354: Add Comma after “conditions”

Line 359: Add Comma after “study”

Line 361: Add Comma after “contrast”

Line 362: Add Full-stop after figure number.

Line 384: Add Comma after “condition”

Line 387: Add parentheses (100 mg/L)

Line 392: Add Comma after “condition”

Line 408: Add Comma after “results”

Line 421: Add “such as” after “enzymes”

Line 423: Revise it “Our results are”

Line 426: Add Parentheses (200 mg/L). You did this mistake repeatedly. Please revise it throughout the manuscript.

Line 429: Add Comma after “study”

1. It is better to polish the conclusion section. It is not up to the mark. Substantially revise this portion with justifications and logical statements.

2. For all the figures: you have missed the Alphabetical Letter within the figures. Please insert the letters to indicate the small figures within the main figures.

3. Secondly, all the small figures are not combined properly into one figure. Please revise it carefully.

Reviewer #2: The present study explored the possibility to using alpha-tocopherol as a mitigating agent against drought s tress on Lens culinaris. The manuscript provides an important set of data demonstrating the beneficial effects of alpha-tocopherol. The experiment is well conducted and data analyzed in an intelligible way. In its current state however, the manuscript is not suitable for publication. Major revision is needed. Substantial language editing is needed to reach standard for publication in this Journal. Long sentences should be avoided as much as possible. Besides, there are many inaccurate statements and/or incomplete information that need to be addressed throughout the manuscript.

Below are detailed comments:

The title of the study can be simplified… for ex. “Effect of exogenous of alpha-tocopherol on physio-biochemical attributes and agronomic performance of Lentil (Lens culinaris Medik.) under drought stress”

The introductive sentence of Abstract can be refined to highlight the link between drought stress and oxidative burst, and the potential mitigating effects of alpha-tocopherol. Further, the strategy used to induce drought stress (L33) can be mentioned in the Abstract.

The Abstract is too descriptive and doesn’t provide insights about specific mechanisms underlying the mitigating effects of alpha-tocopherol. A major overhaul is needed. Notably, it is important to highlight the interrelations of obtained data, and their relative relevance in influencing plant performance under drought stress.

An alternative word can be used for “Chaos” (L31). And throughout the manuscript, the term “environmental Chaos” could be avoided, as it seems quite subjective.

The Introduction section can be improved by underlining the regional biophysical context to highlight how severe is drought stress in Punjab. And the reason for choosing Lentil (Lens culinaris) as biological material in this study should be clearly stated.

There is also a need for more background information to justify selection of Alpha-tocopherol as alleviating agent against drought stress… Existing reports on Tocopherol changes under drought stress, or its other potential association with plant responses to stress could be useful. Information in L383-L387, L389-391 can also be explored for the purpose. This may smooth the formulation of the hypothesis of the study, which is not clear in current manuscript.

The “Methods” part needs to be rewritten in a more rigorous way, by providing necessary information required for possible replication of the study:

- L99-100: What the ratio 2:1 stands for, since physico-chemical analyses suggested that the soil is sandy loam. Please clarify this.

- What is the relative proportion of water and 70 % EtOH in the vehicule solution used for preparation of Alpha-tocopherol treatment?

- What was the basis for selecting the different levels of Tocopherol treatments (100, 200, 300 mg/L) and those of drought treatments (20 and 25d drought periods)

- Please rewrite statement for statistical analyses, and integrate the approach used for correlation analysis.

- Please accurately present Dubois methodology for measurement of soluble total content.

- It is apparent that the experiment was designed with two factors (tocopherol level, level of drought stress). However, authors appear to suggest three factors (L198). Please address this matter.

The quality of graphs is rather poor… Please improve their resolution. And for better readability, the legend of different figures can be modified as (within drought stress period): (i) Control, (ii) Drought + 0 mg/L tocopherol, (iii) Drought + 100 mg/L tocopherol, (iv) Drought + 200 mg/L tocopherol, (v) Drought + 300 mg/L tocopherol. Information about statistical inference, including the interaction of studied factors can also be integrated in legends.

It is important to indicate how strong is the coefficient of correlation (Table 2) rather than only mentioning that the correlation is significant. This way, results may be interpreted in a different way, notably drought effects on chlorophyll (a and b) content…

Besides, to gain more insights about the influence of studied factors, it would be interesting to analyze how studied parameters are related one with another, which would allow a deeper discussion of results.

Overall, the discussion didn’t help understanding the specific mode of action of Tocopherol in attenuating harmful effects of drought stress. Since drought stress is commonly related to stomatal closure, resulting in impaired Photosynthesis efficiency, it would have been interesting to clarify which parameter (s) in the photosynthesis apparatus was/were site(s) of Tocopherol beneficial action…

Following minor flaws could also be addressed:

- L303-306. This section is not at its appropriate place… Please check.

- L378: alternative expression can be found for “distinctly significant”.

- L344: “key metabolic activities”…What are they?

- For better readability, at the first occurrence in each section, please mention what different acronyms stand for.

6. PLOS authors have the option to publish the peer review history of their article (what does this mean?). If published, this will include your full peer review and any attached files.

Reviewer #1: **Yes: **Shahbaz Atta Tung

Reviewer #2: No

---

## [Author Response · Author response to Decision Letter 0]

8 Jun 2021

Author Response to the Reviewer 01 Comments

1. Highlights have been added accordingly.

2. The term “Climatic condition” is replaced with “climatic conditions”

3. The word “ill” is replaced with word “drastic”.

4. The word “induced drought” is revised to “drought induced” throughout the manuscript.

5. Instead of “alpha” abbreviation “α” is used throughout the manuscript. 

6. Comma is added after word “findings”.

7. Comma is added after word “plants”.

8. Comma is added after word “regimes”.

9. Full stop is added after word “Fabaceae”.

10. “Induced drought” is revised to “drought induced” throughout the manuscript.

11. “Potassium (K) available” is revised to “available Potassium (K)”

12. Details of “CVG” added as suggested by reviewer#1.

13. “α” is added before tocopherol.

14. TAB is added at the start of paragraph.

15. Comma is added after word “Treatments”.

16. Full stop is added after word “error” throughout the manuscript for all figures

17. Word “group” has been deleted after word “treatments”.

18. All the figure legends are mentioned together below the references. 

19. Comma is added after word “treatments”

20. The word “condition” has been removed.

21. Comma has been added after “condition”.

22. The word “slow” is correct according to grammar rules. 

23. Comma is added after word “study”.

24. Comma is added after word “contrary”.

25. Parenthesis has been removed after word “in”

26. Comma has been added after word “conditions”

27. Comma has been added after word “study”.

28. Comma has been added after word “contrast”

29. Full stop has been added after (Fig. 4a-d).

30. Comma has been added after word “condition”

31. “100 mg/L” has been written inside parenthesis.

32. Comma has been added after word “condition”

33. Comma has been added after word “results”

34. Word “such as” is added after word “enzymes”

35. Word “were” is revised with word “are”

36. Parenthesis have been added and revised throughout the manuscript.

37. Comma has been added after word “study”

Author Response to the Reviewer 02 Comments

1. Title has been simplified as “Effect of exogenous alpha-tocopherol on physio-biochemical attributes and agronomic 

 performance of Lentil (Lens culinaris Medik.) Under drought stress” 

2. A brief mechanism of mitigating effect of α–tocopherol under drought stress has been added in abstract section by 

 highlighting the link between drought stress and oxidative burst.

3. The strategy used for inducing drought stress has been added in the abstract section.

4. The term “Chaos” has been replaced with “adversities” throughout the manuscript.

5. The required information has been added in the introduction section.

6. Being rich in protein content, highly exportable to international markets and a major cash crop of Pakistan, were the 

 main reasons behind selecting lentil as biological material in the present study.

7. Further information has been added in the abstract, introduction and discussion sections.

8. The “Method” part has been improved, where improvement was needed.

9. The ratio of sand and silt was mistakenly written; it has been removed and replaced with accurate statement “with each 

 pot containing 3kg sandy loam soil”.

10.The relative proportion of water and 70 % EtOH was 9:1 used for preparation of Alpha-tocopherol treatment. 

11. Inferences from petri dish experiment revealed better responses in terms of radicles and plumules length with applied 

 treatments of tocopherol used in pot experiment. Moreover, 20 and 25d drought periods were induced in the original 

 trial because low levels of drought induced periods (5 to 15 d) did not show obvious physio-morphological effects on 

 plant growth in the experiment that was designed

12. Statistical analyses statement has been rewritten and the approach used for correlation analysis has also been added.

13. Methodology for soluble sugar content has been rewritten and presented accurately.

14. The experiment actually consisted of two factors, three factors were written mistakenly, and rectification has been 

 incorporated in the revised manuscript. 

15. Quality of graphs has been improved and the legends of graphs have been modified according to the suggestions given 

 by reviewer#2.

16. A detailed account of tocopherol, in attenuating harmful effects of drought stress and the site where tocopherol shows 

 maximum activity has been mentioned in the discussion section from previous literature.

17. Rectification has been done by arranging the small figures alphabetically within the main figure (figure #7).

18. The term “distinctly” has been removed from results section, as it seems a bit inappropriate.

19. The key metabolic activities are photosynthesis and respiration.

20. Conclusion section has been improved by adding logical arguments.

21. All the acronyms in the manuscript have already been added below the abstract section.

22. Correlation was significant at P≤0.05. The magnitudes of positivity and negativity of coefficient of correlation for 

 different parameters have already been added in Table 2.

---

## [Decision Letter · Decision Letter 1]

6 Jul 2021

PONE-D-21-05714R1

Effect of exogenous alpha-tocopherol on physio-biochemical attributes and agronomic performance of Lentil (Lens culinaris Medik.) under drought stress

PLOS ONE

Dear Dr. Sami,

Thank you for submitting your manuscript to PLOS ONE. After careful consideration, we feel that it has merit but does not fully meet PLOS ONE’s publication criteria as it currently stands. Therefore, we invite you to submit a revised version of the manuscript that addresses the points raised during the review process.

ACADEMIC EDITOR: Dear authors, plz have a look on the comments raised by reviewers. I agree with both reviewers that some minor changes are still required..

We look forward to receiving your revised manuscript.

Kind regards,

Basharat Ali, Ph.D

Academic Editor

PLOS ONE

Journal Requirements:

Reviewers' comments:

Reviewer's Responses to Questions

**Comments to the Author**

1. If the authors have adequately addressed your comments raised in a previous round of review and you feel that this manuscript is now acceptable for publication, you may indicate that here to bypass the “Comments to the Author” section, enter your conflict of interest statement in the “Confidential to Editor” section, and submit your "Accept" recommendation.

Reviewer #1: (No Response)

Reviewer #2: (No Response)

2. Is the manuscript technically sound, and do the data support the conclusions?

Reviewer #1: Yes

Reviewer #2: Yes

3. Has the statistical analysis been performed appropriately and rigorously? 

Reviewer #1: Yes

Reviewer #2: N/A

4. Have the authors made all data underlying the findings in their manuscript fully available?

Reviewer #1: Yes

Reviewer #2: Yes

5. Is the manuscript presented in an intelligible fashion and written in standard English?

Reviewer #1: Yes

Reviewer #2: Yes

6. Review Comments to the Author

Reviewer #1: The authors have made good efforts in revising the manuscript according to the comments. But at some points, I still feel that changes should be made. You should have to follow the below mentioned points before acceptance for publication.

1. Even you have revised your paper according to the suggestions but the newly added information is still the same in the Abstract and Conclusion. So, please be rational and make the information unique for both of these sections. They must not have similar sentences.

2. You have added the highlights and the information is good but please follow the standard rules for highlights. Make the bullet points and divide the information in these bullet points.

3. Please arrange the abbreviations in the footnotes of the first page to make them more obvious and precise.

Reviewer #2: The manuscript has been substantially improved, and authors have satisfactorily addressed previous comments, though some minor concerns persist.

For example, the sentence in paragraph L104-L111 is too long! It can be split in shorter sentences as follows: From [In plant (L104)] to [environment (L106)], from [Among (L104) to oxidative stress (109)], from [It is suggested (L109) to condition (L111)]. Please double-check the manuscript for similar flaws.

Further, I am not sure whether the style for presentation of “Highlights” (added section: L61-77) is conform to the Journal instructions…

7. PLOS authors have the option to publish the peer review history of their article (what does this mean?). If published, this will include your full peer review and any attached files.

Reviewer #1: **Yes: **Shahbaz Atta Tung

Reviewer #2: No

---

## [Author Response · Author response to Decision Letter 1]

14 Jul 2021

Response letter to the Reviewers and Editor's comments has already been attached.

---

## [Editor Report · Decision Letter 2]

19 Jul 2021

Effect of exogenous alpha-tocopherol on physio-biochemical attributes and agronomic performance of Lentil (Lens culinaris Medik.) under drought stress

PONE-D-21-05714R2

Dear Dr. Sami Ullah,

We’re pleased to inform you that your manuscript has been judged scientifically suitable for publication and will be formally accepted for publication once it meets all outstanding technical requirements.

Kind regards,

Basharat Ali, Ph.D

Academic Editor

PLOS ONE
---

## [Editor Report · Acceptance letter]

26 Jul 2021

PONE-D-21-05714R2 

Effect of exogenous alpha-tocopherol on physio-biochemical attributes and agronomic performance of Lentil (*Lens culinaris* Medik.) under drought stress 

Dear Dr. Ullah:

I'm pleased to inform you that your manuscript has been deemed suitable for publication in PLOS ONE. Congratulations! Your manuscript is now with our production department. 

Kind regards, 

on behalf of

Dr. Basharat Ali 

Academic Editor

PLOS ONE